# Partitioning gene-based variance of complex traits by gene score regression

**Wenmin Zhang[1], Si Yi Li[2], Tianyi Liu[2], Yue Li[1,2]***

**1** Quantitative Life Science, McGill University, Montreal, Quebec, Canada, **2** Department of Computer Science, McGill University, Montreal, Quebec, Canada

* yueli@cs.mcgill.ca

**Data Availability Statement:** All of the data used in this paper are described under subsection "Real data application" in the manuscript and pasted below as reference. We applied our approach to investigate pathway enrichment for 27 complex

## Abstract

The majority of genome-wide association studies (GWAS) loci are not annotated to known genes in the human genome, which renders biological interpretations difficult. Transcriptome-wide association studies (TWAS) associate complex traits with genotype-based prediction of gene expression deriving from expression quantitative loci(eQTL) studies, thus improving the interpretability of GWAS findings. However, these results can sometimes suffer from a high false positive rate, because predicted expression of different genes may be highly correlated due to linkage disequilibrium between eQTL. We propose a novel statistical method, Gene Score Regression (GSR), to detect causal gene sets for complex traits while accounting for gene-to-gene correlations. We consider non-causal genes that are highly correlated with the causal genes will also exhibit a high marginal association with the complex trait. Consequently, by regressing on the marginal associations of complex traits with the sum of the gene-to-gene correlations in each gene set, we can assess the amount of variance of the complex traits explained by the predicted expression of the genes in each gene set and identify plausible causal gene sets. GSR can operate either on GWAS summary statistics or observed gene expression. Therefore, it may be widely applied to annotate GWAS results and identify the underlying biological pathways. We demonstrate the high accuracy and computational efficiency of GSR compared to state-of-the-art methods through simulations and real data applications. GSR is openly available at https://github.com/li-lab-mcgill/GSR.

## Introduction

Genome-wide association studies (GWAS) have been broadly successful in associating genetic variants with complex traits and estimating trait heritabilities in large populations [1–4]. Over the past decade, GWAS have quantified the effects of individual genetic variants on hundreds of polygenic phenotypes [5, 6]. GWAS summary statistics have enabled various downstream analyses, including partitioning heritability [7], inferring causal single nucleotide polymorphisms (SNPs) using epigenomic annotations [8], and gene sets enrichment analysis for

traits (Fig 2b) using publicly available summary statistics and genotype-expression weights based on 1,264 GTEx whole blood samples (https://www.gtexportal.org/home/datasets2). The GWAS summary statistics were downloaded from public database https://data.broadinstitute.org/alkesgroup/sumstats_formatted/. We downloaded expression weights and reference LD structure estimated in 1000 Genomes using 489 European individuals, from the TWAS/FUSION website (http://gusevlab.org/projects/fusion/) Franke lab cell-type-specific gene expression dataset were obtained from https://data.broadinstitute.org/mpg/depict/depict_download/tissue_expression. In addition, we applied GSR to test for gene set enrichment in three well-powered types of cancer: breast invasive carcinoma (BRCA, 982 cases and 199 controls), thyroid carcinoma (THCA, 441 cases and 371 controls) and prostate adenocarcinoma (PRAD, 426 cases and 154 controls), using gene expression datasets from The Cancer Genome Atlas (TCGA). Uniformly processed (normalized and batch-effect corrected) gene expression datasets from TCGA and GTEx were obtained from https://figshare.com/articles/Data_record_3/5330593. Gene expression and phenotype were standardized before supplying to the GSR software. Standard GSEA was also performed for comparison. Gene sets were downloaded from the MSigDb website http://software.broadinstitute.org/gsea/msigdb/index.jsp. Here we combined BIOCARTA, KEGG and REACTOME to create a total of 1,050 gene sets. We also downloaded the 4,436 GO biological process terms as additional gene sets as well as the 189 gene sets pertaining to oncogenic signatures for the TCGA data analysis.

**Funding:** The research is supported by Canada First Research Excellence Fund (CFREF) Healthy Brains, Healthy Life (HBHL) New Investigator fund (249591) at McGill University and Mon- treal Neurologic Institute (MNI) and NSERC Discovery Grant (RGPIN-2019-0621). The funders had no role in study design, data collection and analysis, decision to publish, or preparation of the manuscript. No author received a salary from any of the funders.

**Competing interests:** The authors have declared that no competing interests exist.

complex traits [9]. However, it remains challenging to link these genetic associations with known biological mechanisms. One main reason is that the majority of the GWAS loci are not located in known genic regions of the human genome.

Transcriptome-wide association studies (TWAS) [10–12] offer a systematic way to integrate GWAS and the reference genotype-gene expression datasets, such as the Genotype-Tissue Expression project (GTEx) [13], via expression quantitative loci (eQTL). In TWAS, we could first quantify the impact of each genetic variant on expression variability in a population and obtain predicted gene expression levels based on new genotypes; Then, we could correlate the predicted gene expression with the phenotype of interest in order to identify pivotal genes [10]. Moreover, when individual-level genotypes and gene expression levels are not available, we could still quantify gene-to-phenotype association (i.e. TWAS statistics) using only the marginal effect sizes of SNPs on the phenotype and on gene expression respectively [11]. These concepts and implementations have largely facilitated explanation of genetic association findings at the gene or the pathway level.

However, as depicted in Fig 1, TWAS are often confounded by the gene-to-gene correlation of the *genetically predicted* gene expression due to the SNP-to-SNP correlation i.e., linkage disequilibrium (LD) [12]. Consequently, relying on the TWAS statistics may lead to false positive discoveries of causal genes and pathways. One approach to address this problem is to fine-map causal genes by inferring the posterior probabilities of configurations of each gene being causal in a defined GWAS loci and then test gene set enrichment using the credible gene sets of prioritized genes [14]. However, this approach is computationally expensive, restricted to GWAS loci, and sensitive to the arbitrary thresholds used for determining the credible gene set and the maximum number of causal genes per locus.

Another method called PASCAL [9] projects SNP signals onto genes while correcting for LD, and then performs pathway enrichments as the aggregated transformed gene scores, which asymptotically follows a chi-square distribution. However, PASCAL does not leverage the eQTL information for each SNP thereby assuming that *a priori* all SNPs have the same effect on the gene. Stratified LD score regression (LDSC) offers a principle way to partition the SNP heritability into functional categories, defined based on tissue or cell-type specific epigenomic regions [7] or eQTL regions of the genes exhibiting a strong tissue specificity [15]. Although LDSC is able to obtain biologically meaningful tissue-specific enrichments, it operates at the SNP level, rendering it difficult to assess enrichment of gene sets. Moreover, neither PASCAL nor LDSC is able to integrate the observed gene expression data measured in a disease cohort (rather than the reference cohort) that are broadly available across diverse studies of diseases including cancers such as The Cancer Genome Atlas (TCGA) [16].

Although expression-based methods, such as gene set enrichment analysis (GSEA), are often adopted in combination with the observed gene expression and phenotypes [17], they generally do not account for the gene-to-gene correlation. While this type of correlation is usually caused by shared transcriptional regulatory mechanisms across genes, GSEA still likely produces false positives in identifying causal pathways.

In this study, we present a novel and powerful gene-based heritability partitioning method that jointly accounts for gene-to-gene correlation and integrates information captured at either the SNP-to-phenotype or the SNP-to-gene level. We utilize this method to identify plausible causal gene sets or pathways for complex traits. We showcase its high accuracy and computational efficiency in various simulated and real scenarios.

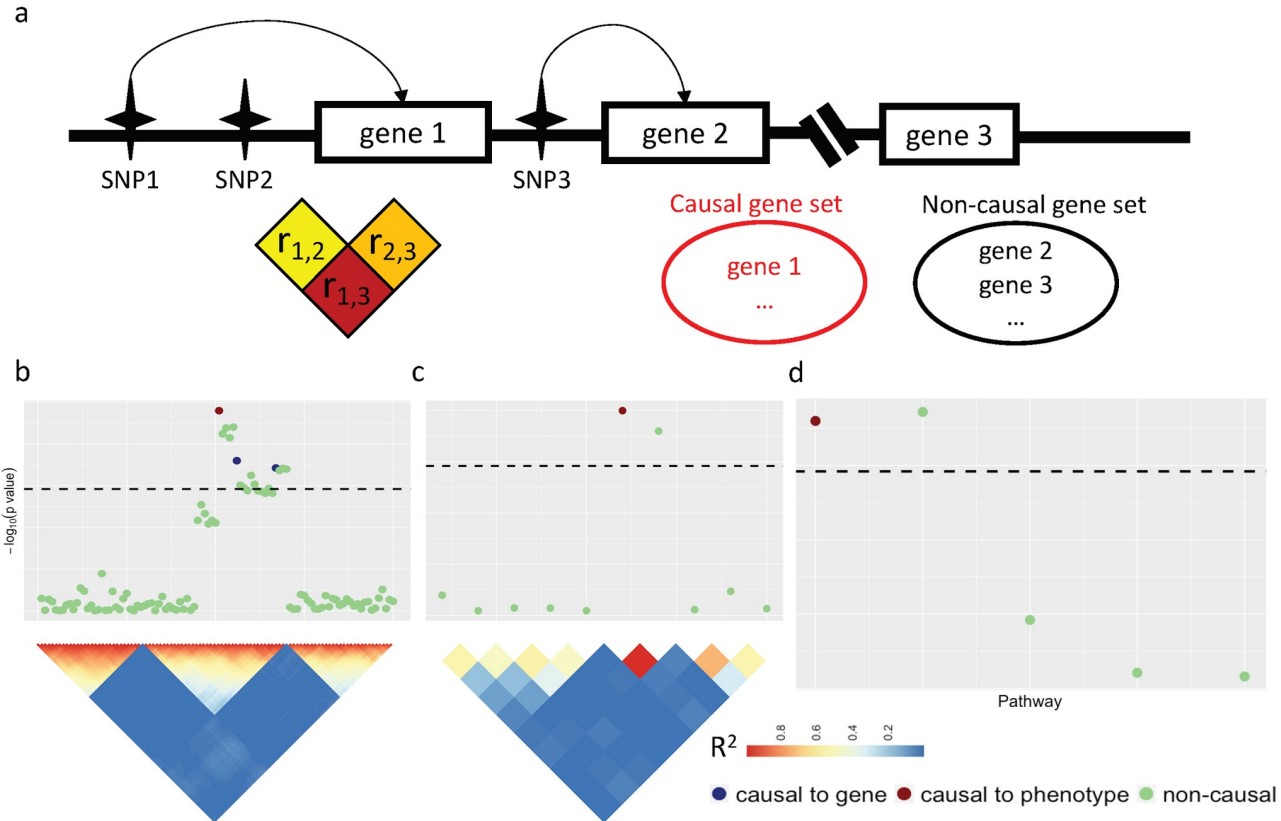

**Fig 1. Overview of confounding effects on pathway analysis.** (a) A hypothetical example illustrates the confounding issue when using the genetically predicted transcriptome to assess the pathway enrichments for a target phenotype. The causal gene set includes a causal gene 1, which is linked to non-causal gene 2 via their respective causal SNPs 1 and 3, which are in strong linkage disequilibrium. (b) A GWAS locus. SNP associations with the target phenotype are summarized. The causal SNP for the causal gene is in red. The SNPs that drive non-causal genes are in blue. The rest of the SNPs are in green. SNPs exhibit correlated signals due to the linkage disequilibrium (LD) as displayed by the upper triangle of the SNP-SNP Pearson correlation matrix; (c) A TWAS locus. The gene-to-gene correlation is partly induced by the SNP-to-SNP correlation and partly due to intrinsic co-regulatory expression program. (d) Pathway associations based on averaged gene associations.

## Methods

### Partitioning gene-based variance of complex traits

We assume gene expression has linearly additive effects on a continuous polygenic trait $y$:

$$y_i = \sum_j A_{ij}\alpha_j + \epsilon_i \tag{1}$$

where $A_{ij}$ denotes the expression of the $j$-th gene in the $i$-th individual for $i \in \{1, \ldots, N\}$ individuals and $j \in \{1, \ldots, G\}$ genes; $\alpha_j$ denotes the **true** effect size of the $j$-th gene on the trait and $\alpha_j \sim N(0, \sigma_j^2)$; $\epsilon_i$ denotes the residual for the $i$-th individual in this linear model and $\epsilon_i \sim N(0, \sigma_\epsilon^2)$.

Here we further assume that both $y$ and $A$ are standardized such that $\frac{1}{N}\sum_i y_i = 0$, $\frac{1}{N}y^\top y = 1$, $\frac{1}{N}\sum_i A_{ij} = 0$ and $\frac{1}{N}A_j^\top A_j = 1$, for $j \in \{1, \ldots, G\}$.

We define the estimated **marginal** effect size of the $j$-th gene on the trait as $\hat{\alpha}_j$:

$$\hat{\alpha}_j \quad = \frac{1}{N} A_j^\top y \tag{2}$$

$$= \frac{1}{N} A_j^\top \left( \sum_k A_k \alpha_k + \epsilon \right) \tag{3}$$

$$= \sum_k \frac{1}{N} A_j^\top A_k \alpha_k + \frac{1}{N} A_j^\top \epsilon \tag{4}$$

$$= \sum_k \hat{r}_{jk} \alpha_k + \epsilon' \tag{5}$$

where $\epsilon' = \frac{1}{N} A_j^\top \epsilon$ with

$$Var(\epsilon') = \frac{1}{N^2} A_j^\top Var(\epsilon) A_j = \frac{1}{N} \sigma_\epsilon^2$$

and $\hat{r}_{jk} = \frac{1}{N} A_j^\top A_k$ is the estimated Pearson correlation in gene expression between the $j$-th gene and the $k$-th gene.

We define $\chi_j^2 = N \hat{\alpha}_j^2$. Then, if we further assume $\alpha$, $r$ and $\epsilon'$ are independent, we have

$$E[\chi_j^2] \quad = E[N \hat{\alpha}_j^2] \tag{6}$$

$$= NE\left[ \left( \sum_k \hat{r}_{jk} \alpha_k + \epsilon' \right)^2 \right] \tag{7}$$

$$= N \sum_k E[\hat{r}_{jk}^2] E[\alpha_k^2] + \sigma_\epsilon^2 \tag{8}$$

Now, consider $C$ gene sets $C_c$, where $c \in \{1, \ldots, C\}$ and denote the proportion of total trait variance explained by the $c$-th gene set as $\tau_c$ with $\tau_c = \frac{\sum_{j \in C_c} Var(\alpha_j)}{|C_c|}$. Here, $|C_c|$ denotes the number of genes in the $c$-th gene set.

Consequently,

$$E[\alpha_k^2] = Var(\alpha_k) = \sum_{c:k \in C_c} \tau_c$$

By approximating $E[\hat{r}_{jk}^2]$ with $\hat{r}_{jk}^2 + \frac{1}{N}$, we have that

$$E[\chi_j^2] \quad = N \sum_k E[\hat{r}_{jk}^2] E[\alpha_k^2] + \sigma_\epsilon^2 \tag{9}$$

$$= N \sum_c \tau_c \sum_{k \in C_c} \hat{r}_{jk}^2 + \sum_c \tau_c + \sigma_\epsilon^2 \tag{10}$$

$$= N \sum_c \tau_c l(j, c) + 1 \tag{11}$$

where we define gene score as $l(j, c) = \sum_{k \in C_c} \hat{r}_{jk}^2$ and $Var(y) = \sum_c \tau_c + \sigma_\epsilon^2 = 1$ since the continuous trait is normalized.

Therefore, if we are able to obtain estimates for $\chi_j^2$ and $C$ gene score $l(j, c)$ for $j \in \{1, \ldots, G\}$ and $c \in \{1, \ldots, C\}$, we will be able to perform linear regression of the estimated $\chi_j^2$ on $l(j, c)$, and derive regression coefficient that is an estimate for each $\tau_c$ ($c \in \{1, \ldots, C\}$), respectively.

These are available from GWAS summary statistics of SNP-to-trait effect sizes, eQTL summary statistics of SNP-to-gene expression effect sizes, and a reference LD panel. Specifically,

1. Suppose we have estimated effect sizes ($\beta_{p \times 1}$) of $p$ SNPs based on a GWAS including $N_{\text{gwas}}$ samples, i.e.

$$\beta = \frac{1}{N_{\text{gwas}}} X^\top y$$

where $X_{N\text{gwas}} \times p$ is the standardized genotype. Meanwhile, we have the eQTL summary statistics $W$ estimated using

$$A_{eQTL} = X_{eQTL} W$$

Therefore, the predicted gene expression in GWAS is given by

$$A = XW$$

Since

$$\chi_j^2 \quad = N\hat{\alpha}_j^2 \tag{12}$$

$$= N(\frac{1}{N} A_j^\top y)^2 \tag{13}$$

$$= N(\frac{1}{N} W_j^\top X^\top y)^2 \tag{14}$$

$$= N(W_j^\top \beta)^2 \tag{15}$$

the required $\chi_j^2$ can be estimated without accessing any individual-level data.

2. Furthermore, a reference LD panel $\Sigma_{p \times p}$ summarizing SNP-to-SNP correlation in the matched population with the GWAS study can provide estimates for $r_{jk}$ as

$$R = [r_{jk}] \tag{16}$$

$$= \frac{1}{N} A^\top A \tag{17}$$

$$= \frac{1}{N} W^\top X^\top X W \tag{18}$$

$$= \frac{1}{N} W^\top \Sigma W \tag{19}$$

It is noteworthy that with individual-level gene expression data, we can also easily obtain the required $\chi_j^2$ and $R = [r_{jk}]$ by definition.

In practice, many gene sets are not disjoint and share common genes with each other. Therefore, we regress one gene set at a time along with a "dummy" gene set that include the union of all of the other genes. The dummy gene set is used to account for unbalanced gene sets and to stabilize estimates of $\tau_c$. We also include an intercept in the regression model to alleviate non-gene-set biases, for example, positive correlation between gene scores and true gene effect sizes that could lead to intercept greater than 1 and negative correlation between gene scores and true gene effect sizes could lead to intercept smaller than 1.

## Simulation design

To assess the accuracy of our GSR approach, we simulated causal SNPs for gene expression as well as causal gene sets for a continuous trait based on real genotypes and known gene sets from existing databases. Our simulation included two stages: At stage 1, we first simulated gene expression based on reference genotype panel. We then estimated SNP-gene effects $\hat{W}_g$ for each gene $g$ based on the simulated gene expression and genotype, which were then used to predict gene expression; At stage 2, separately, we simulated the a continuous trait using simulated gene expression based on genotype, and estimated the marginal SNP-phenotype effects.

**Simulation step 1: simulating gene expression**:

1. To simulate individual genotype, we first partitioned genotype data for 489 individuals of European ancestry obtained from the 1000 Genomes Project [18] into independent 1703 LD blocks as defined by LDetect [19];

2. We then randomly sampled 100 LD blocks and used only those 100 LD blocks for the subsequent simulation; We used 100 LD blocks as opposed to whole genome to reduce computational burden required for multiple simulation runs;

3. For LD block $j$ ($j \in \{1, \ldots, 100\}$) of an individual $i$ ($i \in \{1, \ldots, 500\}$), we randomly sampled from the 489 available samples for block $j$, and concatenated these sampled LD blocks $1, \ldots, 100$ for this individual. We repeated this procedure to simulate genotype $\mathbf{X}_{\text{ref}}$ for $N_{\text{ref}} = 500$ individuals as a reference population;

4. We standardized the simulated genotype $\mathbf{X}_{\text{ref}}$;

5. We randomly sampled $k$ in-cis causal SNPs per gene within $\pm 500$ kb around the gene, where $k = 1$ (default). We also experimented different number of causal SNPs $k \in \{2, 3, \text{all in-cis SNPs}\}$;

6. We sampled SNP-gene weights $\mathbf{W}_g \sim \mathcal{N}(0, h_g^2/k)$ where gene expression heritability $h_g^2 = 0.1$ (default), which is the variance of gene expression explained by genotype. We also experimented different gene heritability $h_g^2 = \{0.2, 0.3, 0.4, 0.5\}$;

7. We then simulated gene expression $\mathbf{A}_{g,ref} = \mathbf{X}_{ref} \mathbf{W}_g + \epsilon$, where $\epsilon \sim \mathcal{N}(0, \sigma_\epsilon^2)$ and $\sigma_\epsilon^2 = \frac{1}{N_{\text{ref}}} \| \mathbf{X}_{\text{ref}} \mathbf{W}_g \|^2 (\frac{1}{h_g^2} - 1) \mathbf{I}_{N_{\text{ref}}}$ to match the desired heritability: $\frac{1-h_g^2}{h_g^2} = \frac{\sigma_\epsilon^2}{\|\mathbf{X}_{\text{ref}} \mathbf{W}_g\|^2 / N_{\text{ref}}}$

8. Finally, we applied LASSO regression $\mathbf{A}_{g,\text{ref}} \sim \bar{\mathbf{X}} \mathbf{W}_g$ to get $\hat{\mathbf{W}}_g$ for each gene.

**Simulation step 2: simulating phenotype**:

1. We simulated another $N_{gwas}$ = 50,000 GWAS individuals by the 100 predefined LD blocks among the 489 Europeans in 1000 Genome data, following the same procedures as decribed above;

2. We then standardized the simulated genotype $\mathbf{X}_{gwas}$;

3. We then sampled a causal pathway $\mathcal{C}_c$ from MSigDB such that all of the $G_c \equiv |\mathcal{C}_c|$ genes in $\mathcal{C}_c$ were causal genes for the phenotype;

4. For each non-causal pathway, we removed genes that were also present in the causal pathway. We removed non-causal pathways containing fewer than five genes afterwards (default); Alternatively, in more realistic scenarios, we allowed for sharing genes with causal pathways by non-causal pathways;

5. We sampled gene-phenotype effect $\alpha \sim \mathcal{N}(0, \sigma_\alpha^2/G_c \mathbf{I}_{G_c})$, where the phenotypic variance explained by gene expression $\sigma_\alpha^2 = 0.1$ (default). We also experimented different $\sigma_\alpha^2 \in \{0.1, 0.2, 0.3, 0.4, 0.5\}$;

6. We simulated gene expression $\mathbf{A}_c$ as in step 1 for the $N_{gwas}$ individuals, and standardized it to obtain $\bar{\mathbf{A}}_c$

7. We simulated a continuous trait using causal gene expression: $\mathbf{y} = \bar{\mathbf{A}}_c \boldsymbol{\alpha} + \epsilon_y$ where $\epsilon_y \sim \mathcal{N}(0, \sigma_{\epsilon_y}^2)$. Here, $\sigma_{\epsilon_y}^2 = \frac{1}{N_{gwas}} \parallel \bar{\mathbf{A}}_c \boldsymbol{\alpha} \parallel^2 (\frac{1}{\sigma_\alpha^2} - 1) \mathbf{I}_{N_{gwas}}$ to match the predefined proportion of variance explained: $\frac{1-\sigma_\alpha^2}{\sigma_\alpha^2} = \frac{\sigma_{\epsilon_y}^2}{\parallel \bar{\mathbf{A}}_c \boldsymbol{\alpha} \parallel^2 / N_{gwas}}$

8. Lastly, we computed GWAS summary SNP-to-trait effect size: $\beta = \frac{1}{N} \mathbf{X}_{gwas}^\top \mathbf{y}$

We repeated these simulation procedures 100 times. Unless otherwise stated, while we were experimenting various settings, we kept the other settings at their default values: $k$ = 1 causal SNP per gene; gene expression variance explained per causal SNP $h_g^2 = 0.1/k$; phenotypic variance explained per gene $\sigma_\alpha^2 = 0.1$; one causal pathway. Using these obtained summary statistics, we were able to perform GSR, PASCAL, LDSC and FOCUS in each simulated scenario.

**Applying existing methods.** **PASCAL**: PASCAL was downloaded from https://www2.unil.ch/cbg/index.php?title=Pascal [9]. We executed the software using default settings. **LDSC**: Stratified LD score regression software was downloaded from https://github.com/bulik/ldsc [15]. Because LDSC operates on SNP level, we considered SNPs located within ± 500 kb around genes in each pathway to be involved in the corresponding pathway. Then, for each pathway, we computed the LD scores over all chromosomes. We experimented the options of running LDSC with and without the 53 baseline annotations using our simulated data. We found that LDSC running without the 53 baseline worked better in our case. One possible reason is that the baseline annotations cover genome-wide SNPs whereas there are much fewer SNPs in the simulated pathways. **FOCUS**: We downloaded FOCUS [14] from https://github.com/bogdanlab/focus. We used FOCUS to infer the posterior probability of each gene being causal for the phenotype across all of the LD blocks. We then took the 90% credible gene set as follows. We first summed all of the posteriors over all of the genes. We then sorted the genes by the decreasing order of their FOCUS-posteriors. We kept adding the top ranked the gene into the 90% credible gene until the sum of their posteriors was greater than or equal to the 90% of the total sum of posteriors. We used the 90% credible gene set for hypergeometric test for each pathway to compute the p-values. We also tried other thresholds for credible sets ranging from 75% (including the fewest genes) to 99% (including the most genes). **GSEA**: GSEA software was obtained from http://software.broadinstitute.org/gsea [17]. We used the

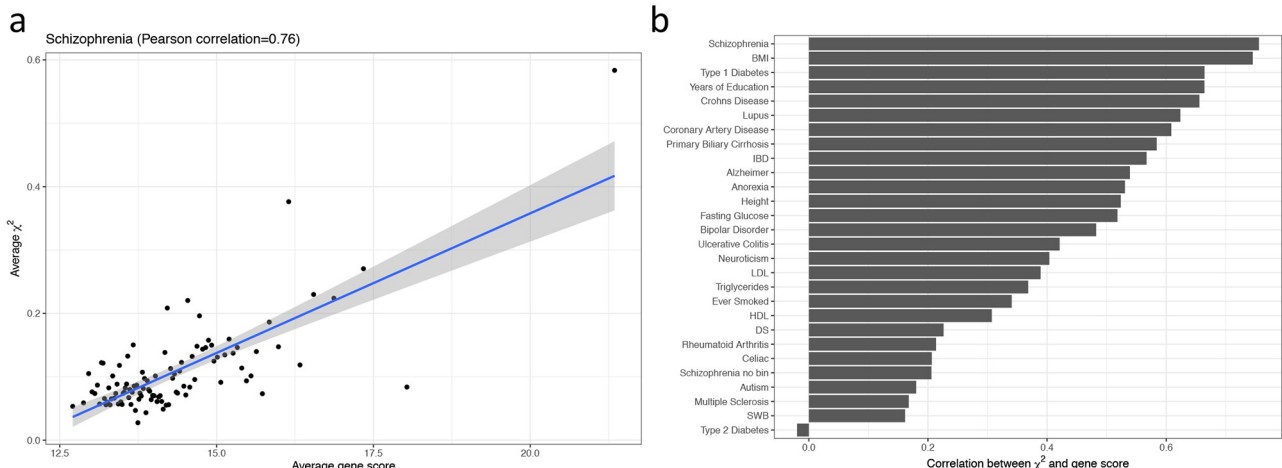

**Fig 2. Gene scores correlated with marginal TWAS summary statistics.** (a) Gene scores were correlated with marginal TWAS chi-square statistics of Schizophrenia. We grouped genes into 100 bins by their gene scores to reduce noise. For each bin, we calculated the average gene scores and chi-square statistics ($\chi^2$). An unbinned version is supplied in **S1 Fig** in S1 File. (b) Summary of Pearson correlation between marginal TWAS summary statistics and gene scores for 27 traits. The negative correlation for T2D indicates that this trait, possibly due to complicated genetic architecture and confounding gene-to-environment interaction and drug effects, is not suitable for using our approach.

command-line version of GSEA to test for gene set enrichments using the observed gene expression and phenotype data.

**Real data application.** We applied our approach to investigate pathway enrichment for 27 complex traits (Fig 2b) using publicly available summary statistics and genotype-expression weights based on 1,264 GTEx whole blood samples. The GWAS summary statistics were downloaded from public database https://data.broadinstitute.org/alkesgroup/sumstats_formatted/ [7]. We downloaded expression weights and reference LD structure estimated in 1000 Genomes using 489 European individuals, from the TWAS/FUSION website (http://gusevlab.org/projects/fusion/) [11, 18]. Franke lab cell-type-specific gene expression dataset were obtained from https://data.broadinstitute.org/mpg/depict/depict_download/tissue_expression.

In addition, we applied GSR to test for gene set enrichment in three well-powered types of cancer: breast invasive carcinoma (BRCA, 982 cases and 199 controls), thyroid carcinoma (THCA, 441 cases and 371 controls) and prostate adenocarcinoma (PRAD, 426 cases and 154 controls), using gene expression datasets from The Cancer Genome Atlas (TCGA). Uniformly processed (normalized and batch-effect corrected) gene expression datasets from TCGA and GTEx were obtained from https://figshare.com/articles/Data_record_3/5330593 [20]. Gene expression and phenotype were standardized before supplying to the GSR software. Standard GSEA was also performed for comparison.

Gene sets were downloaded from the MSigDb website http://software.broadinstitute.org/gsea/msigdb/index.jsp. Here we combined BIOCARTA, KEGG and REACTOME to create a total of 1,050 gene sets. We also downloaded the 4,436 GO biological process terms as additional gene sets as well as the 189 gene sets pertaining to oncogenic signatures for the TCGA data analysis.

## Results

### Gene scores were correlated with TWAS statistics in polygenic complex traits

Our method GSR is built on the hypothesis that the marginal gene effect sizes on the phenotype should be positively correlated with the sum of correlation with other genes, which

include causal genes. To validate this hypothesis, we defined *gene score* for each gene as the sum of its squared Pearson correlation with all of the other genes, derived from gene expression levels. We calculated TWAS marginal statistics as the product of GWAS summary statistics (*β*) and eQTL weights (*W*) derived from the GTEx whole blood samples (Eq 15). To assess the impact of gene-to-gene correlation on TWAS statistic, we correlated the gene scores with the TWAS marginal statistics for 27 complex traits. Overall, most traits had Pearson correlation between the gene score and the marginal TWAS statistic above 0.4. For instance, the correlation in schizophrenia was 0.76 (Inter-Quartile Range: 0.66—0.81 based on 1,000 permutations; Fig 2). This implies a pervasive confounding impact on the downstream analysis, including gene set or pathway enrichment analysis, causal gene identification, etc., using the TWAS summary statistics while assuming independence of genes (Fig 1).

## GSR improved pathway enrichment power

In simulated scenarios with default settings (Methods), compared to PASCAL and LDSC, GSR demonstrated hugely improved computational efficiency (Table 1), superior sensitivity in detecting causal pathways with an improved statistical power as well as competitive specificity in controlling for false positives (Fig 3). Specifically, in 100 simulations, GSR achieved an overall area under the precision-recall curve (AUPRC) of 0.925, and identified the true causal pathway as the most significant one 93 times, compared to 56 times by PASCAL, which only achieved an overall AUPRC of 0.260. Notably, the FOCUS-predicted 75%, 90%, 99% credible gene sets were also significantly enriched for causal pathways (Fig 3).

We then varied four different settings: (a) the number of causal SNP per gene; (2) SNP-gene heritabilities; (3) gene-phenotype variance explained; (4) overlapping causal pathway. We focused our comparison with PASCAL because it directly tested for pathway enrichment and has been demonstrated to outperform other relevant enrichment methods [9]. In all simulation settings, GSR demonstrated an improved power in detecting the causal pathways (**S2 Fig** in S1 File), as it was able to detect causal pathways when multiple SNPs influenced gene expression, when the proportion of variance explained by the gene expression was low, or when the causal and non-causal pathways were allowed to overlap. In contrast, a lot of causal pathways were not deemed significant by PASCAL based on a p-value threshold of 0.001, which was equivalent to a Bonferroni-corrected p-value threshold of 0.1 after correcting for multiple testing on approximately 100 pathways tested per simulation.

## Improved power in pathway enrichment leveraging observed gene expression

One unique feature of GSR is the ability to run not on only the summary statistics but also on observed gene expression, where the gene-gene expression correlation is directly estimated

**Table 1. Comparison of existing methods with GSR.**

| Method | GWAS | TWAS | Measured expression | Running time |
|---|---|---|---|---|
| PASCAL [9] | sum. stat. [*] | | | 10 m |
| LDSC [15] | sum. stat. | | | >24 h [†] |
| FOCUS [14] | sum. stat. | sum. stat. | | >24 h |
| GSEA [17] | | | individual expression | 10 m |
| GSR | sum. stat. | | individual expression | 3 min |

[*] Summary statistics

[†] For custom gene sets, the main computation time for LDSC is calculating the LD score for all of the 1000 Genome SNPs.

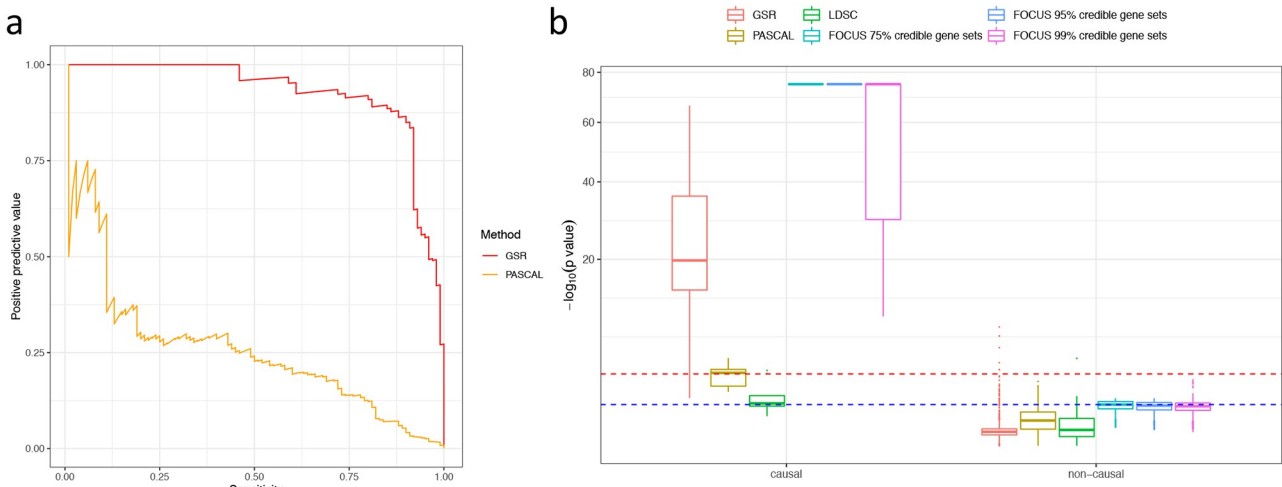

**Fig 3. Evaluation of power and robustness of GSR in detecting causal pathways.** (a) Precision-recall curves for GSR and PASCAL summarizing results from 100 simulations. (b) Summary of p-values obtained by running GSR along with PASCAL, LDSC and FOCUS 10 times. For each method, the enrichment significance for causal pathways and non-causal pathways are displayed. We experimented FOCUS with 75%, 90%, and 99% credible sets for the pathway enrichments. For the ease of comparison, we plotted the y-axis on a square-root negative logarithmic scale. Red line denotes p-value threshold of 0.001; Blue line denotes p-value threshold of 0.1.

from the in-sample gene expression. To evaluate the accuracy of this application, we simulated gene expression and phenotype for 1,000 individuals, which were provided as input to GSR for pathway enrichment analysis. As a comparison, we applied GSR to the summary statistics generated from the same dataset.

As in the simulation above, the SNP-expression weights were estimated from a separate set of 500 reference individuals whereas the SNP-phenotype associations were estimated from only 1,000 individuals. Notably, the sample size for the GWAS cohort is much smaller than the previous application to mimic the real data where usually fewer than 1000 individuals have both the RNA-seq and phenotype available (e.g., TCGA). Additionally, we applied standard GSEA [17] to the same dataset with the observed gene expression. We observed an improved power of GSR when using the observed gene expression over GSR using the summary statistics (Fig 4), whereas GSEA had a comparable performance as the latter. Specifically, all causal pathways in the simulated replicates had a p-value below 0.001, with the largest p-value being $7.5 \times 10^{-6}$, as determined by GSR using observed gene expression, while no causal pathway reached this level of significance (with the smallest p-value being $1.4 \times 10^{-2}$) determined by GSEA. We also compared the performances of GSR using observed gene expression to GSEA in various simulation settings and obtained consistent conclusions (**S3 Fig** in S1 File).

## Gene set enrichments in complex traits

Applying GSR to 27 complex traits, we revealed various pathways where the enriched gene sets were biologically meaningful. For example, the enriched gene sets for high density lipoprotein (HDL) predominantly involve lipid metabolism; In contrast, for Lupus, gene sets were enriched in interferon signalling pathways, a known immunological hallmark. We listed the top 10 enrichments over gene sets from MSigDB and Gene Ontology terms for HDL and the autoimmune trait Lupus in **S1 Table** in S1 File.

Additionally, we applied GSR to test cell-type-specific enrichments using 205 cell types, 48 of which were derived from GTEx and 157 cell types were derived from Franke lab datasets [15]. We observed biologically meaningful cell type-specific enrichment for the 27 complex

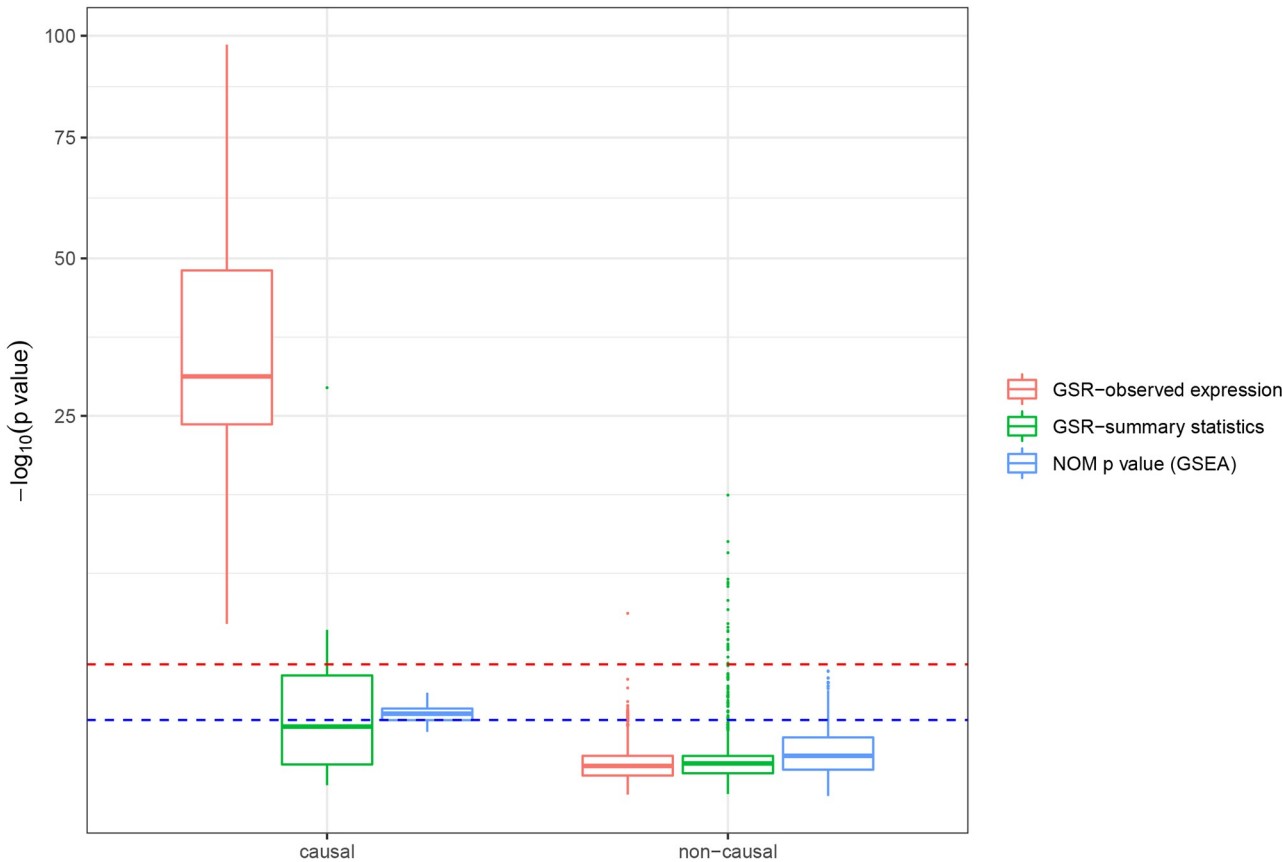

**Fig 4. Comparison of pathway enrichment determined by GSR using or not using observed gene expression information, and by GSEA.** Nominal (NOM) p-values yielded by GSEA were summarized. Red line denotes p-value threshold of 0.001; Blue line denotes p-value threshold of 0.1.

traits (Fig 5). In particular, central neural system cell-specific gene sets were enriched for schizophrenia, immune cell-specific gene sets for lupus, immune cell-specific and digestive tract cell-specific gene sets for Crohn's disease and cardiac cell-specific gene sets for coronary artery disease. Lastly, we correlated traits based on their gene set enrichments and observed meaningful phenotypic clusters, suggesting shared biological mechanisms by the related phenotypes (**S4 Fig** in S1 File). For example, Crohn's disease and ulcerative colitis, two subtypes of inflammatory bowel disease formed a cluster; Neurological diseases, schizophrenia and bipolar disorder formed a cluster; Moreover, lipid traits including LDL, HDL, and Triglycerides formed their own cluster.

## Application on observed gene expression

Lastly, using expression profiles of BRCA, THCA and PRAD from TCGA and GTEx [20], we tested the enrichments of 186 oncogenic gene sets as well as 1,050 gene sets from BIOCARTA, KEGG, and REACTOME in each type of tumor. Overall, we observed a significantly stronger enrichments for the oncogenic signatures with higher p values compared to the more general gene sets across all three tumour types (t-test p-value = $6.4 \times 10^{-25}$, $9.0 \times 10^{-29}$ and $1.1 \times 10^{-23}$ for BRCA, PRAD and THCA respectively; **S5 Fig** in S1 File). As a comparison, we also ran standard GSEA and observed qualitatively similar enrichments (**S5 Fig** in S1 File).

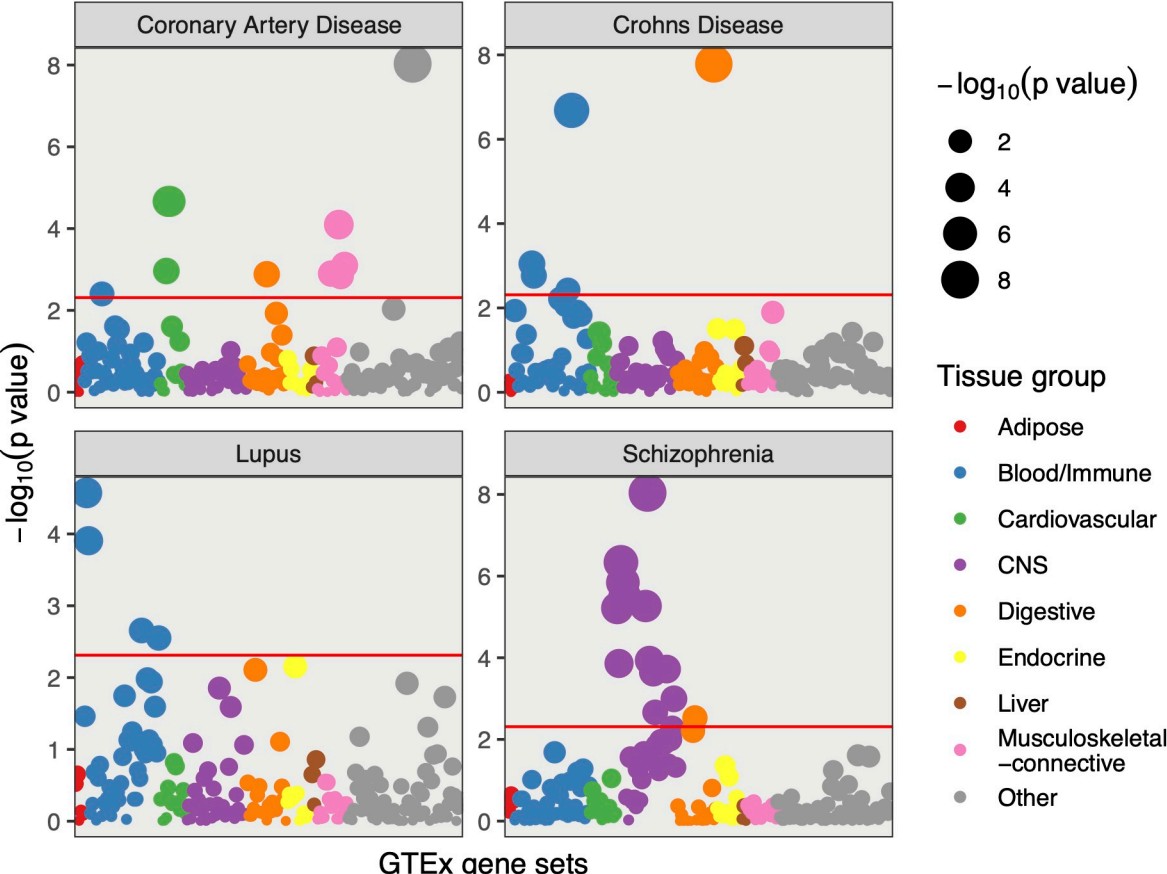

**Fig 5. Cell-type-specific enrichment of gene sets for representative complex traits.** GSR was applied to each complex trait in order to identify significantly enriched gene sets among 205 pre-defined cell-type-specific gene sets, represented by nine different colors. Gene sets were indicated by dots and were aligned in the same order on the x-axis. Red lines indecate Bonferroni-corrected p-value threshold (0.05).

## Discussion

In this work, we describe GSR, an efficient method to test for gene set or pathway enrichments using either GWAS summary statistics or observed gene expression and phenotype information. We demonstrate robust and powerful detection of causal pathways in extensive simulation using our proposed method compared to several state-of-the-art methods. When applying to the real data, we also obtained biologically meaningful enrichments of relevant gene sets and pathways. These features warrant GSR a widely applicable method in various study settings with an aim to interpret association test results and capture the underlying biological mechanisms.

Our approach has superior computational efficiency. In particular, GSR took only 3-5 minutes running on the full summary statistics and less than 5 minutes on the full gene expression data with one million SNPs and 20,000 genes to test for enrichments of over 4,000 gene sets. In our simulations, it is not surprising that FOCUS can accurately fine-map causal genes as the simulation designs followed similar assumptions adopted by FOCUS [14]. However, FOCUS is at least 20 times slower than GSR. For the simulated data, FOCUS took 30 minutes to fine-map all of the genes in GWAS loci whereas GSR took under three minutes to test for pathway enrichments on the same machine. Additionally, the computational cost of FOCUS is exponential to the number of causal genes considered within each locus whereas GSR is not affected

by the number of causal genes. Also, because GSR operates at genome-wide level, no threshold is needed to decide which genes to be included whereas FOCUS needs user-defined threshold for constructing the credible gene set for the subsequent hypergeometric enrichment test. Given these advantages, we envision that GSR will be a valuable tool for the bioinformatic community and statistical genetic community as a fast way to investigate the functional implications of complex polygenic traits.

In different simulation settings, GSR exhibits improved pathway enrichment power over PASCAL and LDSC, two popular methods for partitioning heritability and identifying causal gene sets. Since GSR leverages SNP-to-gene association summaried by eQTL weights while either PASCAL or LDSC operates on the SNP level, without considering this intermediate association, such improvements are expected and beneficial. Given that existing eQTL studies have yielded reliable estimates of SNP-to-gene effects and are easily accessible, we consider GSR more promising in bridging the gap between large GWAS and multi-faceted functional annotations on the genome.

One unique feature of our approach is that it could leverage the observed individual-level gene expression that are broadly available to calculate more accurate in-sample gene-gene correlation. Indeed, we observed more accurate detection of causal pathway for modest sample size (1000 individuals) where the phenotype and gene expression are available compared to GSR operating only on summary statistics. In real data analysis, we demonstrate that GSR can achieve similar biologically meaningful enrichments as GSEA when applied to the observed gene expression. On the other hand, GSR has the advantage of working with summary statistics when the individual gene expression and phenotype are not available where GSEA could hardly be performed.

It is noteworthy that p values generated by different methods in this study are not directly comparable due to different model assumptions, statistical tests being used, sampling methods, etc. However, we posit that the p-values themselves are informative in reality. When gene set enrichment analysis is performed in related studies, p-values are usually directly adopted to identify specific signals as a common practice. Therefore, GSR may be promising to refine interpretation and reveal under-identified biological mechanisms in existing studies, as it is able to yield smaller p-values for the true underlying pathways.

Our method has important limitations. First of all, our method relies on pre-computed eQTL weights, which might absorb measurement uncertainty, confounding effects as well as stochastic errors. Besides, it is usually unknown how these weights vary across different populations, i.e. whether the effect of each SNP on the corresponding gene expression is conserved, particularly when investigation is carried out on a diseased population while using a non-diseased reference population. Furthermore, our method is built on an important assumption that the effect sizes of genes on the trait and the derived gene scores are independent. In practice, if this assumption is violated, our method might suffer from the bias introduced. While no method exists to examine the validity of these properties to our knowledge, since we obtained consistent results in our real data analyses, we posit that our method should be robust in identifying causal pathways. We propose our method could be widely utilized various studies where further calibration of the exact estimates of effect sizes should continuously improve its performance.

## Supporting information

**S1 File.**
(PDF)

## Acknowledgments

We thank Mathieu Blanchette for the helpful comments on the manuscript.

## Author Contributions

**Conceptualization:** Yue Li.

**Investigation:** Yue Li.

**Methodology:** Wenmin Zhang, Yue Li.

**Software:** Si Yi Li, Tianyi Liu, Yue Li.

**Supervision:** Yue Li.

**Writing – original draft:** Yue Li.

**Writing – review & editing:** Yue Li.

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
