## [Decision Letter · Decision Letter 0]

29 Apr 2020

PONE-D-20-07689

Partitioning gene-based variance of complex traits by gene score regression

PLOS ONE

Dear Dr. Li,

Thank you for submitting your manuscript to PLOS ONE. After careful consideration, we feel that it has merit but does not fully meet PLOS ONE’s publication criteria as it currently stands. Therefore, we invite you to submit a revised version of the manuscript that addresses the points raised during the review process.

The expert reviewers provide valuable advice.  Please carefully address each and every concern. It will substantially improve your report.

We would appreciate receiving your revised manuscript by Jun 13 2020 11:59PM. To enhance the reproducibility of your results, we recommend that if applicable you deposit your laboratory protocols in protocols.io, where a protocol can be assigned its own identifier (DOI) such that it can be cited independently in the future. For instructions see: http://journals.plos.org/plosone/s/submission-guidelines#loc-laboratory-protocols

We look forward to receiving your revised manuscript.

Kind regards,

F. Alex Feltus, Ph.D.

Academic Editor

PLOS ONE

Journal Requirements:

'The research is supported by Canada First Research Excellence Fund (CFREF) Healthy

Brains, Healthy Life (HBHL) New Investigator fund (249591) at McGill University and Montreal

Neurologic Institute (MNI) and NSERC Discovery Grant (RGPIN-2019-0621).'

'The funders had no role in study design, data collection and analysis, decision to

publish, or preparation of the manuscript.'

4. Please include a copy of Table 1 which you refer to in your text on page 14.

Additional Editor Comments (if provided):

Reviewers' comments:

Reviewer's Responses to Questions

**Comments to the Author**

1. Is the manuscript technically sound, and do the data support the conclusions?

Reviewer #1: Partly

Reviewer #2: Yes

2. Has the statistical analysis been performed appropriately and rigorously? 

Reviewer #1: No

Reviewer #2: Yes

3. Have the authors made all data underlying the findings in their manuscript fully available?

Reviewer #1: Yes

Reviewer #2: Yes

4. Is the manuscript presented in an intelligible fashion and written in standard English?

Reviewer #1: Yes

Reviewer #2: Yes

5. Review Comments to the Author

Reviewer #1: The manuscript describes an issue when gene set enrichments are performed using genes identified from transcriptome wide association study (TWAS). In TWAS while using expression quantitative trait loci (eQTL) to genetically predict gene expression in a GWAS cohort and performing associations between gene expression and phenotype, genes that are not relevant to the phenotype but are regulated by SNPs in high LD with the causal SNP can obtain high test statistics. This can lead to false discoveries in gene set enrichment analyses. This is a reasonable issue and a valid aim for the study. To address this issue, the authors’ strategy is to regress out the sum of gene-gene correlation from the genes’ marginal statistic and estimate the amount of phenotypic variance explained by the predicted expression of the genes. I found the main text to be sparsely written and confusing in various sections and many aspects of this study are unclear to me. Some analysis approaches are also puzzling to me. Either there are issues with the methodology and/or the procedures could be described in a considerably better way to engage a wide readership. I have the following comments:

1. Section 4.1 line “We calculated TWAS marginal statistic as the product of GWAS summary statistic and eQTL weights derived from the GTEx whole blood samples” is unclear in what was the summary statistic used - effect size? P value? Why was TWAS statistic defined this way when the authors could have used some existing TWAS studies and calculate correlation??

2. Why did the genes have to be binned and the correlation calculated on the average scores? The Fig 2 titles “Correlation between marginal statistic and gene scores 27 traits” are then misleading because it’s actually averaging within bins. To make this analysis more robust, permutations should be performed to assess the significance of correlation. Also, how come the x axis that shows the bins, goes only from 12 to 20 when it should start from 1? The straightforward way would be to calculate the correlation between the gene scores with the TWAS effect size. The authors should comment on why their specific approach was taken.

3. Fig 3: The methods compared all have intrinsically different algorithms, assumptions, statistical tests, number of samplings etc. Are the p values from all these really comparable? Fig 3 legend says “the enrichment score for causal pathways and non-causal pathways” which suggests that a metric such as effect size/fold change would be presented. A fair comparison would include some sort of precision/recall metrics. The authors also ran only 10 simulations which seems quite low, and might explain why some of the interquartile ranges in fig 3 are so large.

4. The method FOCUS seems to perform better in the simulations but the reasoning against using that is that it took 30 mins vs GSR took 3 mins. This is an insufficient argument in favor of GSR, users would definitely prefer accuracy over little extra computational resources.

5. Fig 5 labels are not legible and quite distracting. It is unclear what this figure is really trying to highlight, the relevant pathways come up from the compared methods as well. Only the scale of p values change. Are there relevant pathways that other methods miss but are identified by GSR?

6. Fig 6: This figure is also very briefly explained in the text. What is the x axis and what do multiple points for the same color represent? The other methods are not compared at this point?

7. The figure legends in the manuscript in general are very short and non-informative.

Other comments:

Fig 1C - The number labels in this panel are confusing as this is a hypothetical example. Maybe just lable gene 1, gene 2 etc.

The simulation analysis in Fig 3 is barely explained in the main text and just references the methods. This analysis could be set up in a more informative way in the main text to benefit the readers.

Discussion could be elaborated a little.

Reviewer #2: In this manuscript, the authors proposed a method, Gene Score Regression (GSR), to estimate the phenotypic variance explained by the gene expression and can be used to test for gene set or pathway enrichments based on GWAS summary statistics or the observed gene expression. They performed simulation experiments and also applied GSR to real data. The results supported that GSR is powerful and robust. My major concern is power and false positive rate. In simulation study, the authors only performed 10 times for each setting, if computation time is not a problem, it would be good to perform at least 1000 times to estimate power and false positive rate for some settings, particularly low heritability. Minor concern is the format. It seems that the authors use another format in the beginning but didn’t completely match PLOS ONE’s format. It really confused me when I reviewed this manuscript, so I think the authors should reorganize it. Figure quality and errors should also be careful. I listed my questions as following by section: (please see attachment)

6. PLOS authors have the option to publish the peer review history of their article (what does this mean?). If published, this will include your full peer review and any attached files.

Reviewer #1: No

Reviewer #2: No

---

## [Author Response · Author response to Decision Letter 0]

18 Jun 2020

Reviewer #1

Reviewer #1: The manuscript describes an issue when gene set enrichments are performed using genes identified from transcriptome wide association study (TWAS). In TWAS while using expression quantitative trait loci (eQTL) to genetically predict gene expression in a GWAS cohort and performing associations between gene expression and phenotype, genes that are not relevant to the phenotype but are regulated by SNPs in high LD with the causal SNP can obtain high test statistics. This can lead to false discoveries in gene set enrichment analyses. This is a reasonable issue and a valid aim for the study. To address this issue, the authors’ strategy is to regress out the sum of gene-gene correlation from the genes’ marginal statistic and estimate the amount of phenotypic variance explained by the predicted expression of the genes. I found the main text to be sparsely written and confusing in various sections and many aspects of this study are unclear to me. Some analysis approaches are also puzzling to me. Either there are issues with the methodology and/or the procedures could be described in a considerably better way to engage a wide readership. I have the following comments:

 Thank you for your comments. We have made extensive modifications and clarifications throughout the manuscript. Please also see our response below.

1. Section 4.1 line “We calculated TWAS marginal statistic as the product of GWAS summary statistic and eQTL weights derived from the GTEx whole blood samples” is unclear in what was the summary statistic used - effect size? P value? Why was TWAS statistic defined this way when the authors could have used some existing TWAS studies and calculate correlation??

 We used the effect sizes from both GWAS summary statistics and GTEx eQTL summary statistics. We sought to have a unified way to integrate both GWAS results and eQTL information and we found the product of SNP-to-trait effect sizes in GWAS and SNP-to-gene expression effect sizes could be a proxy for the marginal effect size of the gene on the trait. We have now clarified this in Eq.12-15 and explained in more detail in the Methods in lines 216.

2. Why did the genes have to be binned and the correlation calculated on the average scores? The Fig 2 titles “Correlation between marginal statistic and gene scores 27 traits” are then misleading because it’s actually averaging within bins. To make this analysis more robust, permutations should be performed to assess the significance of correlation. Also, how come the x axis that shows the bins, goes only from 12 to 20 when it should start from 1? The straightforward way would be to calculate the correlation between the gene scores with the TWAS effect size. The authors should comment on why their specific approach was taken.

 The genes were binned in Figure 2 to reduce noise because gene scores, calculated based on eQTL summary statistics may contain inflated noise. This approach was also adopted by the LD-score regression study (Bulik-Sullivan et al. Nature Genetics). However, for comparison, we have added a Supplementary Figure S1 which does not bin gene scores. From there, we observed that the slopes fitted in two cases were very similar, despite the Pearson correlation dropped if the genes were not binned, which indicates the noise inflation. We have provided this information and corrected the figure legend and annotate x axis as average gene score of each bin instead of the bin indicator. In addition, we have also performed 1000 permutations to derive a confidence interval (line 221) for the Pearson correlation estimate, by randomly sampling genes and recreating bins. This CI overlapped with and was centered around the original correlation estimate, thus verifies the robustness of the approach.

3. Fig 3: The methods compared all have intrinsically different algorithms, assumptions, statistical tests, number of samplings etc. Are the p values from all these really comparable? Fig 3 legend says “the enrichment score for causal pathways and non-causal pathways” which suggests that a metric such as effect size/fold change would be presented. A fair comparison would include some sort of precision/recall metrics. The authors also ran only 10 simulations which seems quite low, and might explain why some of the interquartile ranges in fig 3 are so large.

 We completely agree that the p-values obtained from these different algorithms were not directly comparable, as you pointed out. However, we posit that the p-values themselves are very important in practice, because when people perform gene set / pathway enrichment analysis, the ultimate goal, usually, would be to identify some enriched signals. Under the circumstances, it is the p-values that the researchers would need to rely on to pick up these targets. Thus, if an algorithm is able to yield smaller p-values for the true underlying pathways regardless the difference in the underlying null distribution, we consider it would be more useful. We added this consideration to the Discussion in lines 342-348. Nevertheless, we have also provided precision-recall curves (PRCs) and area under PRC measures for GSR and PASCAL, which is most relevant method to ours, based on 100 simulations in Figure 3a, and demonstrated the superiority of our proposed method. 

 We also opted to keep the original figure where we showed 10 simulations for all the other methods (LDSC and FOCUS with different credible gene sets), as (1) it is computationally expensive to apply all methods and (2) based on only 10 simulations we were already able to exemplify the performance of them, with quite narrow interquartile ranges for FOCUS and LDSC.

4. The method FOCUS seems to perform better in the simulations but the reasoning against using that is that it took 30 mins vs GSR took 3 mins. This is an insufficient argument in favor of GSR, users would definitely prefer accuracy over little extra computational resources.

 In our simulations, it is not surprising that FOCUS can accurately fine-map causal genes as the simulation designs followed similar assumptions adopted by FOCUS (Mancuso et al., Nature Genetics). If these assumptions do not hold (which is unknown in real settings), it remains debatable whether FOCUS could still accurately capture all the true signals. It is noteworthy that the computational cost of FOCUS is exponential to the number of causal genes considered within each locus whereas GSR is not affected by the number of causal genes. Also, because GSR operates at genome-wide level, no threshold is needed to decide which GWAS/TWAS loci or which genes to be included whereas FOCUS needs user-defined threshold for choosing those GWAS/TWAS loci and for constructing the credible gene set for the subsequent hypergeometric enrichment test. Taken together, we still posit that GSR still add value is a valuable tool to the relevant TWAS studies given its flexibility to use different sources and increased computational efficiency. We have added these to the Discussion in lines 309-312.

5. Fig 5 labels are not legible and quite distracting. It is unclear what this figure is really trying to highlight, the relevant pathways come up from the compared methods as well. Only the scale of p values change. Are there relevant pathways that other methods miss but are identified by GSR?

 We agree. We have now removed this figure and re-iterated this part of results in lines 272-274.

6. Fig 6: This figure is also very briefly explained in the text. What is the x axis and what do multiple points for the same color represent? The other methods are not compared at this point?

 We have now expanded our explanation in the legend of (currently) Figure 5. Gene sets were indicated by dots and were aligned in the same order on the x-axis and multiple points for the same color represent 9 tissue group. Because this section was mainly for demonstration of the biological interpretation one could get from running GSR, we did not opt to compare to the other methods, which have already been widely used. 

7. The figure legends in the manuscript in general are very short and non-informative.

 Thank you. We have revised all the legends and hopefully they are now more informative.

Other comments:

Fig 1C - The number labels in this panel are confusing as this is a hypothetical example. Maybe just lable gene 1, gene 2 etc.

 We have made adjustment.

The simulation analysis in Fig 3 is barely explained in the main text and just references the methods. This analysis could be set up in a more informative way in the main text to benefit the readers.

 We have added more details to the Methods in section “applying existing methods”.

Discussion could be elaborated a little.

 We have added more discussion upon the utility and limitations of our method.

Reviewer #2

In this manuscript, the authors proposed a method, Gene Score Regression (GSR), to estimate the phenotypic variance explained by the gene expression and can be used to test for gene set or pathway enrichments based on GWAS summary statistics or the observed gene expression. They performed simulation experiments and also applied GSR to real data. The results supported that GSR is powerful and robust. My major concern is power and false positive rate. In simulation study, the authors only performed 10 times for each setting, if computation time is not a problem, it would be good to perform at least 1000 times to estimate power and false positive rate for some settings, particularly low heritability. 

 Thank you for your comments. While we agree that the small number of replications may introduce some uncertainty in ascertaining the power of different methods, we have to admit that the excessively high computational cost did circumscribe our efforts to perform more experiments, taking into account the more time-consuming process in generating complete SNP-gene-phenotype datasets. To this end, we have run GSR and PASCAL, our major competitor, in 100 simulations respectively and updated our results in Figure 3. However, since in previous analyses we found FOCUS consistently gave accurate results (but with exceedingly long time) while LDSC was not specifically built for this type of task, we refrained from applying these two algorithms and added discussions in lines 324-329.

Minor concern is the format. It seems that the authors use another format in the beginning but didn’t completely match PLOS ONE’s format. It really confused me when I reviewed this manuscript, so I think the authors should reorganize it. Figure quality and errors should also be careful. 

 We have reformatted the manuscript and we hope it is more clear now.

I listed my questions as following by section:

1 Introduction

 “In TWAS, we can regress on the expression changes using the genotype information from the reference cohort…” In this description, the dependent variable is the expression changes and the independent variable is the genotype information. According to the cited reference 10, Figure 2 and Equation 1, the dependent variable is the expression, not the expression changes. Please clarify.

 We have corrected this expression in line 15-16. Indeed, the dependent variable is the gene expression.

 In Figure 1 (a), please specific which SNP is 1, 2 and 3, respectively? In (b), the blue SNPs are causal for a non-causal gene. Are the two blue SNPs the most significant SNPs among two non-causal genes, respectively? In (c), what does the number represent? 

 We have revised Figure 1a to specify the SNPs; In Figure 1b the two blue SNPs are not necessarily the most significant ones (in detecting SNP-phenotype associations), as this is merely a hypothetical example; In reality, the causal SNPs for non-causal genes may have larger or smaller p-values, depending on the exact magnitude of linkage to the true causal SNPs for the phenotype; We have removed the misleading numbers in Figure 1c.

2. Related Methods

 “…it does not account for the gene-gene correlation, which is distinct from TWAS-induced correlation but is rather due to the sharing of transcriptional regulatory network among genes.” Please explain again that what leads to TWAS-induced correlation? (LD or other factors)

 TWAS-induced correlation mostly comes from LD. We have re-written this section into the Introduction and specified in lines 24-26.

3. Methods

 Phenotypic variance explained by gene expression

 For Eq (2), does it still hold for binary outcome (y), i.e., logistic regression? 

 We have clarified in the Methods in lines 60-61. This approach can be generalized to binary traits on a liability scale, as has been done in the LD-score regression study (Bulik-Sullivan et al. Nature Genetics).

 Please verify Eq (4) and (5) for the term of A_g^gwas y and A_g y. For OLS solution, they will 〖(A_g^gwas)〗^T y.and A_g^T y, respectively.

 Thank you very much. We have corrected them.

 The gene expression of GWAS (A_g^gwas) were estimated based on a reference panel. This estimation is reliable for “controls” in GWAS that represent generation population as those from the reference panel. Is it still reliable for “cases” in GWAS using estimated values (W ^_g) from the reference panel. 

 We have added this point in the discussion part in line 353-355. Our assumption is that genes instead of genotype directly affect the disease status. In this sense, we think as long as the genetic structure of the two populations match and the bias introduced by population stratification is well controlled, which is normally the case for GWAS study, it is valid to use the same set of weights.

 Please define the notions for i and j. 

 We have defined them in line 63.

 The Eq (6) was not used in the following text. What is the purpose to show β ^ in the Eq(6)? 

 We intended to related it to the GWAS SNP-to-trait effect size. Now we have clarified in lines 83-84.

 “From (8) to (9), we assume that all of the random variables are independent. The assumption holds if gene causal effects are independent and are also independent from the gene-gene correlation.” If assumption doesn’t hold, what kind of bias will be introduced (e.g., underestimate or overestimate)? 

 We have added discussion on the bias in lines 97-100 and lines 356-358.

 Please explain how to obtain χ_g^2. Does it the test statistic from the regression of phenotype y on gene expression (or predicted gene expression)? 

 Now we have clarified in Eq. 12-15.

3.2 Partitioning variance component by gene sets

 “The full derivation is similar to that for Eq (12) and detailed in Supplementary Methods.” Please indicate which section in Supplementary Methods. 

 We have now unified the derivation for both summary statistics and individual level data. All materials have been incorporated into the Methods.

 “Therefore, we regress one gene set at a time along with a dummy gene set that include the union of all of the genes in the gene sets. The dummy gene set is used to account for unbalanced gene sets.” In a regression model for one gene set, does it include two independent variables, i.e., one is the gene score for a given gene set and the other is the gene score for a dummy gene set? In the dummy gene set, does it include genes belonging to the gene set of interest? Why a dummy gene set can be used to account for unbalanced gene sets? Please explain.

 Yes. We have two independent variables in the example you provided. We have further clarified in lines 93-95. The including of a dummy gene set because all genes need to be contained in the our model. 

 According to the main equation, an intercept in a regression model would be close to 1. In what condition, the intercept will be away from 1? In the text, “We also include an intercept in the regression model to properly control non-gene-set biases.” How to control non-gene-set biases? 

- If the intercept is away from 1, one can examine multiple plausible reasons, including such as various forms of interactions, measurement error, correlation between gene scores and effect sizes in Equation 9, etc. It should be noted that our method is built on an important assumption that the effect sizes of genes on the trait and the derived gene scores are independent. In practice, if this assumption is violated, our method might suffer from the bias introduced. For example, positive correlation between gene scores and true gene effect sizes that could lead to intercept greater than 1 and negative correlation between gene scores and true gene effect sizes could lead to intercept smaller than 1. Here, to be exact, we think there would be no easy solution to control for these biases, but an intercept might alleviate such effects if they are additive to the original offset. We have revised this argument in lines 97-100 and added discussion to the Discussion in lines 356-358.

3.3 Gene score regression on total gene expression

 In page 5, α ^_g=1/N_gwas A ^_g^T y, but here α ^_g=A_g^T y. Is it correct without a term of 1/N_real ?

 Yes. We have clarified it now.

 “If one gene is a causal gene and the other is not, we will see inflated summary statistic for the non-causal gene, thereby confounding the detection for causal pathways.”. Does it indicate that the non-causal gene will be detected and the false positive rate then increases in this case?

- We have re-phrased this entire section which might be misleading.

3.4 Simulation

Simulation step 1: simulate gene expression:

 In 1000 Genomes Project, there are 503 individuals of European ancestry. What are the exclusion criteria to remove individuals? And, how many independent blocks were generated?

 There are only 489 individuals of European ancestry from the 1000 Genomes Project that were documented in the TWAS/FUSION project (Gusev, et al., Nature Genetics), which was our data source. We have specified this in lines 190-191. We sampled 100 LD blocks from a total of 1,703 LD blocks determined by LDetect to reduce computational burden. This is now clarified in lines 112-113.

 Genotype was standardized in the reference panel, i.e. 489 individuals, or after simulation in 500 individuals. Please clarify.

 We standardized the genotype after simulation. This is now clarified in lines 121.

 Was a bootstrap technique used to simulate 500 individuals from 489 Europeans in 1000 Genome data?

 We have rephrased the simulation process in lines 117-120 such that this is more clear. We sampled real LD blocks from these 489 individuals and concatenated them. Therefore, the simulated genotypes would consist of LD blocks from different individuals.

 We randomly sampled k in-cis causal SNPs per gene within ±500 kb around the gene, where k = 1 (default).” If k >1, are the randomly sampled k causal SNPs per gene independent (LD r2<0.2)?

- These k SNPs did not have to be independent; no LD threshold was imposed because we posit this would better preserve the LD structure.

3.5 Data sets and 3.6 Running existing methods

 Different methods require different data type (such as summary statistics or individual genotype/expression data for SNP or gene level) to perform analysis. Please indicate what data type that these five methods in Table 1 require and what datasets that they used for analysis.

 We have added this to Table 1.

 Please provide the sample size information of dataset that were used in this manuscript, including TWAS, GWAS, TCGA and GTEx.

- We have now provided the sample sizes accordingly in lines 186, 195-197.

4. Results

4.1 Gene scores correlate with TWAS statistics in polygenic complex traits

 Please reorganize the 4.1 section. Some parts of description should be presented in the method section.

 We have reorganized this section.

 “We calculated TWAS marginal statistic as the product of GWAS summary statistic and eQTL weights derived from the GTEx whole blood samples.” This sentence indicates how to calculate TWAS marginal statistic for each SNP. However, GSR was proposed for gene level analysis, so please explain how to calculate TWAS marginal statistic for a given gene, e.g. summation of the products of GWAS summary statistic and eQTL weights within a given gene.

 This refers to the current Equation 15, where W would be the eQTL weights and beta would be GWAS summary statistics. This has been clarified in lines 215-216.

 “This implies a pervasive confounding impacts on the downstream analysis using the TWAS statistic (Figure 1e) when using existing approaches that mostly assume independence of genes.” Please provide some examples for downstream analysis. And, Figure 1e should be Figure 2e

 Thank you. We have expanded this sentence in lines 222-224 and corrected the figure reference.

 For Figure 2 (a), please explain why gene bin was used to show the correlation? Why not directly use gene score and chi squared? For (b), the correlation of gene score and TWAS marginal statistic is negative for T2D. How to interpret the negative correlation and what confounders could lead to the negative correlation? Figure panel names doesn’t match to the description in the figure legend.

- The genes were binned in Figure 2 to reduce noise because gene scores, calculated based on TWAS summary statistics may contain inflated noise. This approach was also adopted by the LD-score regression study (Bulik-Sullivan et al. Nature Genetics). However, for comparison, we have added a Supplementary Figure S1 which does not bin gene scores. From there, we observed that the slopes fitted in two cases were very similar, despite the Pearson correlation dropped if the genes were not binned, which indicates the noise inflation. In Figure 2b, the negative correlation for T2D indicates that this trait, possibly due to complicated genetic architecture and confounding gene-to-environment interaction and drug effects, is not suitable for using our approach. We therefore later illustrated the utility of our method using traits with higher correlation, such as schizophrenia. We have also corrected the panel labels.

4.2 GSR improves pathway enrichment power

 In Table 1, what does “Exprs” refer?

 That referred to observed gene expression. We have spelled it out and added information to the legend.

 In Figure 3, please indicate what blue and red dotted lines represent. For y axis, the label doesn’t match the description in the figure legend. Here, is the “enrichment score” (in the description) shown as the p-value?

 Thank you. We have added and corrected information to the figure legend.

 “Notably,… enrichment test.” This whole paragraph should be moved to discussion section, since there is a section called “Discussion and Conclusion”.

- We have rephrased and re-organized accordingly.

4.3 Improved power in pathway enrichment when using the observed gene expression

 “To evaluate the accuracy of this application, we simulated gene expression and phenotype for 1000 individuals, which were provided as input to GSR for pathway enrichment analysis.” Is it another simulation study other than section 4.2? If yes, please describe the procedures in the simulation section.

- Simulation of gene expression is now described in the Methods “Simulation step 2”.This would then not involve using reference TWAS summary statistics.

4.4 Gene set enrichments in complex traits

 In this section, the description related to Materials and Methods should be reorganized.

 We have re-organized this section.

 In Figure 5, it is very unclear to directly put pathway names onto the main figure area, e.g. a band on the bottom misleading the number of gene set to test. On the right tail of Figure 5, i.e., significant for FOCUS, not for GSR, it shows that some gene set/pathway enrichments were detected by FOCUS, but not by GSR. Does it imply that GSR is not powerful in some cases? If yes, please try to evaluate this limitation.

 We have removed this Figure which could be misleading and re-organized this section.

 For cell-type-specific enrichment analyses, was W ^_g estimated from gene expressions of specific cell type?

- No, the cell-type-specific enrichment analyses only utilized cell-type-specific gene sets identified in GTEx and Franke lab datasets; the weights were not specifically estimated for each type of cell, but was based on TWAS using GTEx whole blood samples. This is specified in lines 216-217. Thus, they do not directly represent cell-type-specific gene expression. We have also discussed this limitation in the Discussion in lines 350-355.

4.5 Application on observed gene expression

 Please specify sample size for each cancer, including case and control, in the observed gene expression analyses.

 We have now specified in the Methods in lines 195-197.

Supplementary information

 There are duplications in Supplementary information and Methods. Please reorganize.

 Thank you. We have removed the duplicated sections.

---

## [Decision Letter · Decision Letter 1]

22 Jul 2020

PONE-D-20-07689R1

Partitioning gene-based variance of complex traits by gene score regression

PLOS ONE

Dear Dr. Li,

Thank you for submitting your manuscript to PLOS ONE. After careful consideration, we feel that it has merit but does not fully meet PLOS ONE’s publication criteria as it currently stands. Therefore, we invite you to submit a revised version of the manuscript that addresses the points raised during the review process.

Please address the **minor comments **from Reviewer #2 and take the opportunity to deep read once more before acceptance.

We look forward to receiving your revised manuscript.

Kind regards,

F. Alex Feltus, Ph.D.

Academic Editor

PLOS ONE

Reviewers' comments:

Reviewer's Responses to Questions

**Comments to the Author**

1. If the authors have adequately addressed your comments raised in a previous round of review and you feel that this manuscript is now acceptable for publication, you may indicate that here to bypass the “Comments to the Author” section, enter your conflict of interest statement in the “Confidential to Editor” section, and submit your "Accept" recommendation.

Reviewer #1: All comments have been addressed

Reviewer #2: All comments have been addressed

2. Is the manuscript technically sound, and do the data support the conclusions?

Reviewer #1: Yes

Reviewer #2: Yes

3. Has the statistical analysis been performed appropriately and rigorously? 

Reviewer #1: Yes

Reviewer #2: Yes

4. Have the authors made all data underlying the findings in their manuscript fully available?

Reviewer #1: (No Response)

Reviewer #2: Yes

5. Is the manuscript presented in an intelligible fashion and written in standard English?

Reviewer #1: Yes

Reviewer #2: Yes

6. Review Comments to the Author

Reviewer #1: The manuscript text edits and the clarifications provided by the authors have addressed my comments.

Reviewer #2: The authors clarified my questions and reorganized well. I just have four minor questions as follows. Please see attachment.

7. PLOS authors have the option to publish the peer review history of their article (what does this mean?). If published, this will include your full peer review and any attached files.

Reviewer #1: No

Reviewer #2: No

---

## [Author Response · Author response to Decision Letter 1]

28 Jul 2020

Reviewer #2

The authors clarified my questions and reorganized well. I just have four minor questions as follows.

- Thank you for your feedback.

Methods

Partitioning gene-based variance of complex traits

1. “…,we will be able to perform linear regression and derive regression coefficient that is an estimate for each τc, respectively.” Please indicate what dependent and independent variables are for the linear regression.

- We have indicated now in line 79. We regressed Chi-squares (dependent variable) on gene scores (independent variables).

Simulation design

2. In simulation step 1.3, for 500 individuals, did you sample each block for each individual with replacement? For example, for individual i (I = 1, …, 500) and block j (j = 1, …, 100), you randomly sampled one block j from 489 blocks of j. And, you concatenated sampled block 1, …, 100 for individual i. Please provide a clear description.

- For LD block j (j in {1,...,100}) of an individual i, we randomly sampled from the 489 available samples for block j, and concatenated these sampled LD blocks 1,...,100 for this individual. We repeated this procedure to simulate genotype X_ref for N_ref = 500 individuals as a reference population

- We have further clarified this in lines 118-121.

Results

3. For Figure 2, I think it would be good to add the explanation of “how to interpret negative correlation for T2D in the figure description.

- The negative correlation for T2D indicates that this trait, possibly due to complicated genetic architecture and confounding gene-to-environment interaction and drug effects, is not suitable for using our approach.

- We have added the rationale in the figure legend.

4. In Table 1, “*Summary statistics” and “†For custom gene sets,…” are footnotes and should not be put in the table title.

- Thank you. We have moved those to the footnotes.

---

## [Editor Report · Decision Letter 2]

31 Jul 2020

Partitioning gene-based variance of complex traits by gene score regression

PONE-D-20-07689R2

Dear Dr. Li,

We’re pleased to inform you that your manuscript has been judged scientifically suitable for publication and will be formally accepted for publication once it meets all outstanding technical requirements.

Kind regards,

F. Alex Feltus, Ph.D.

Academic Editor

PLOS ONE
---

## [Editor Report · Acceptance letter]

6 Aug 2020

PONE-D-20-07689R2 

Partitioning gene-based variance of complex traits by gene score regression 

Dear Dr. Li:

I'm pleased to inform you that your manuscript has been deemed suitable for publication in PLOS ONE. Congratulations! Your manuscript is now with our production department. 

Kind regards, 

on behalf of

Dr. F. Alex Feltus 

Academic Editor

PLOS ONE